# Cell-Type-Specific Profiling of the *Arabidopsis thaliana* Membrane Protein-Encoding Genes

**DOI:** 10.3390/membranes12090874

**Published:** 2022-09-10

**Authors:** Sergio Alan Cervantes-Pérez, Marc Libault

**Affiliations:** 1Department of Agronomy and Horticulture, Center for Plant Science Innovation, University of Nebraska-Lincoln, Lincoln, NE 68503, USA; 2Single Cell Genomics Core Facility, Center for Biotechnology, University of Nebraska-Lincoln, Lincoln, NE 68588, USA

**Keywords:** *Arabidopsis thaliana*, plasma membrane proteins, single-cell transcriptomic, root cells

## Abstract

Membrane proteins work in large complexes to perceive and transduce external signals and to trigger a cellular response leading to the adaptation of the cells to their environment. Biochemical assays have been extensively used to reveal the interaction between membrane proteins. However, such analyses do not reveal the unique and complex composition of the membrane proteins of the different plant cell types. Here, we conducted a comprehensive analysis of the expression of Arabidopsis membrane proteins in the different cell types composing the root. Specifically, we analyzed the expression of genes encoding membrane proteins interacting in large complexes. We found that the transcriptional profiles of membrane protein-encoding genes differ between Arabidopsis root cell types. This result suggests that different cell types are characterized by specific sets of plasma membrane proteins, which are likely a reflection of their unique biological functions and interactions. To further explore the complexity of the Arabidopsis root cell membrane proteomes, we conducted a co-expression analysis of genes encoding interacting membrane proteins. This study confirmed previously reported interactions between membrane proteins, suggesting that the co-expression of genes at the single cell-type level can be used to support protein network predictions.

## 1. Introduction

Membrane proteins play a central role in the perception and subsequent transduction of biological signals. These signals control cell-to-cell communication and the response of plant cells to environmental stimuli. To recognize, and eventually translate, complex stimuli into molecular and cellular responses, membrane proteins often belong to large protein complexes that are organized in unique compartments (e.g., plasma membrane micro-domains [1,2,3]). Characterizing these complexes is important to better understand the response of plant cells to their environment.

The characterization of protein–protein interactions requires the implementation of molecular, cellular, and in silico assays [4,5]. To date, yeast two-hybrid assays, and microscopy-based fluorescent imaging technologies [e.g., bimolecular fluorescence complementation (BiFC), split luciferase, split fluorescent proteins, or fluorescence resonance energy transfer (FRET)] have been extensively used to test protein–protein interactions. Several limitations exist when implementing these in vivo methods. First, they are often conducted in heterologous biological systems (i.e., yeast or heterologous plant systems). Second, these methods require the engineering of the proteins to include additional amino acid sequences (e.g., split fluorescent proteins, DNA-binding, and activation domains) that might interfere with the interaction of the proteins. Third, these assays require the co-expression of the two protein partners in the same cell, presuming that the two proteins tested are present in the same cell at the same time.

As an alternative strategy, biochemical assays, such as affinity purification and size exclusion chromatography of protein complexes, followed by the characterization of the peptidic sequences by mass-spectrometry, are also broadly used to characterize the interactions between proteins. For instance, focusing on the root of the model plant *Arabidopsis thaliana*, Gilbert and Schulze applied size exclusion chromatography to reveal the interaction between 1752 pairs of membrane proteins [6].

Interacting proteins pre-suppose their presence in the same cell and at the same time. Considering the half-life of proteins and their encoding transcripts, we assume that the genes encoding interacting proteins should be active in the same cell or cell type. This statement suggests that the two genes encoding the interacting proteins must be co-expressed in the cell. Alternatively, a slight shift in the timing of gene transcription, combined with the differential half-life of the transcripts, might also result in the presence of the two protein partners in the same cell. Finally, when considering the expression of genes in different cell and cell types, two interacting proteins could be co-localized in the same cell upon cell-to-cell trafficking, notably through the plasmodesmata [7]. 

The single-cell transcriptomic approach now opens new avenues for estimating the unique use of genes between different cell types composing the plants, and will support the high-resolution of gene co-expression [8,9,10,11,12,13,14,15,16,17,18,19,20,21,22,23,24,25,26,27,28]. Here, we describe the use of single-cell resolution transcriptomic datasets to reveal the cell-type-specific activity of membrane protein-encoding genes in the Arabidopsis root. Taking advantage of the recent release of plant single-cell (sc) and single-nucleus (sNuc) RNA-seq datasets from the Arabidopsis root [9,28], we analyzed the activity of Arabidopsis genes encoding interacting membrane proteins in the root. Our results confirmed that the genes encoding interacting membrane proteins were co-expressed in the same clusters; and at a similar level of expression. We also found that the different cell types composing the Arabidopsis roots are characterized by unique transcriptional profiles of the 965 membrane protein-encoding genes, suggesting that the Arabidopsis root cell types are characterized by specific membrane proteomes.

## 2. Materials and Methods

### 2.1. Data Acquisition, Preprocessing, and Processing of the Transcriptomes

The fastq files of publicly available scRNA-seq and sNucRNA-seq were obtained from the NCBI (SRA) repository. For this study, we used the raw dataset from two different publications for *Arabidopsis thaliana* roots growing under the same conditions, but prepared differently, depending on the protocol for each method. For the scRNA-seq, we download the dataset under the accession number GSE123013 and for sNucRNA-seq, under the number GSE155304. Five scRNA-seq and three sNucRNA-seq libraries were processed individually using the 10X Genomics Cell Ranger program v6.1.1.0 (https://support.10xgenomics.com/single-cell-gene-expression/software/downloads, accessed on 8 September 2022; Lincoln, NE, USA) using 10X Genomics Cell Ranger count to align with the reference genome/annotation of Arabidopsis from TAIR10 (https://www.arabidopsis.org/download/index-auto.jsp%3Fdir%3D%252Fdownload_files%252FGenes%252FTAIR10_genome_release; last accessed: 21 July 2022). Alternatively, we generated a specific annotation file for membrane proteins to align the reads against these specific annotations.

Following this step, we applied a cleaning step using SoupX v1.6.0 (https://github.com/constantAmateur/SoupX, accessed on 8 September 2022) to remove the background contamination [29], and doublets filtering using DoubletDetection (https://github.com/JonathanShor/DoubletDetection, accessed on 8 September 2022) [30]. Finally, a statistical threshold of the data distribution in the interval of confidence of 95% was applied to remove the outliers. We used Seurat version 4 for the processing of the transcriptomes [31]. Integration anchors were chosen for the combined set of five scRNA-seq and three sNucRNA-seq datasets using the first 20 dimensions of the canonical correlation analysis method. To reduce the complexity of the datasets, a dimensionality reduction using Uniform Manifold Approximation and Projection (UMAP) was performed. For downstream analysis, the expression values were obtained separately for the subset of nuclei belonging to each cluster cell type using the AverageExpression method from the Seurat program v4.0.6 (https://github.com/satijalab/seurat/, accessed on 8 September 2022).

### 2.2. Functional Annotation of the Cell Clusters

To assign a cell type for each cluster of the Arabidopsis root (scRNA-seq and sNucRNA-seq), we selected the 101 cell type gene markers for Arabidopsis root used in [28]. For the visualization of the gene markers, we created a dot plot using the .rds file generated for this study. Alternatively, we used a mixed method to annotate the cell clusters of specific membrane protein root UMAP, first selecting the barcodes for each cell cluster and second, using the highly specific genes in the root membrane UMAP to find the specific expression in the whole root UMAP.

### 2.3. Correlation Analysis of Transcriptomic Profiles

To verify the co-expression of previously tested protein membranes [6], we designed a correlation analysis regarding a specific number of 1752 pairs of membrane proteins tested in the above-mentioned study. The design started with the transcriptome datasets of sNucRNA-seq and scRNA-seq for the entire diversity of cell types. Using Seurat, we extracted the average expression of each cell cluster for each dataset. After that, we applied the filtering of expression, with a minimum of 0.15. In parallel, we applied the same filtering for a control dataset using random expression of random gene pairs. Finally, we applied a Pearson’s correlation for each dataset and the control dataset. The random list of genes was generated in Python v 3.6.9 (https://www.python.org/downloads/release/python-369/, accessed on 8 September 2022), and the data management was realized in MySQL v 14.14 Distrib 5.7.36 (Austin, TX, USA) for Linux.

## 3. Results and Discussion

### 3.1. Enhanced Clustering Analysis of the Arabidopsis Root Nuclei According to Their Transcriptomic Profiles

To conduct a comprehensive analysis of the expression patterns of the Arabidopsis genes encoding interacting membrane proteins in the root, we first revisited the analysis of the Arabidopsis root sc- and sNucRNA-seq datasets [28] by applying SoupX, a bioinformatic package capable of removing ambient transcripts from sc/sNuc RNA-seq datasets [29]. The inclusion of this additional computational step led to the generation of updated UMAPs composed of 12,207 cells and 9689 nuclei, divided into 18 and 16 cell and nucleus clusters, respectively (Figure 1A,B).

We found medians of 4041 and 1441 expressed genes per cell and per nucleus, respectively (versus 4739 and 1124 expressed genes per cell and nucleus [28]), for a total of 24,494 and 23,501 expressed genes in the Arabidopsis root cells and nuclei, respectively (versus a total of 25,177 and 24,510 expressed genes from isolated cells and nuclei, respectively). To annotate the cellular and nuclear clusters (Figure 1A,B), we first analyzed the activity of previously characterized 101 Arabidopsis root cell-type and cell-death marker genes [28] (Figure 1C). Overall, upon annotation, the clusters were organized per cell type, as previously reported [28]. We use this updated resource to analyze the transcriptional profile of membrane protein-encoding genes.

### 3.2. Correlation Analysis between Gene Co-Expression and Protein Interactome

The 965 membrane protein-encoding genes composing the Arabidopsis root interactome selected for this study have been characterized based on the interaction between their encoded proteins. Specifically, 1752 pairs of membrane proteins have been previously characterized based on their high-confidence interactions by applying size-exclusion chromatography and mass spectrometry [6]. A pre-requirement for two proteins to interact is that they be present in the same cell at the same time. With the exception of the translocation of proteins between cells, we hypothesize that the genes encoding interacting proteins should be co-expressed in the same cell or cell type. This hypothesis also presupposes that the half-life of transcripts and proteins does not drastically differ between the genes.

To verify this hypothesis and further support the interaction between membrane proteins [6], we analyzed the co-expression of genes encoding interacting proteins at the single-cell level for the 1752 pairs of high-confidence interactomes (Appendix A). To do so, we mined both the protoplast and nucleus UMAPs (Figure 1). When considering the scRNA-seq cluster, we found 516 (cluster A; 29.5% of the 1752 pairs) and 1099 (cluster G; 62.7%) pairs with co-expressed genes (Figure 2A). Taking all the 18 protoplast clusters together, the genes of 1456 pairs were found to be expressed in at least one cluster. Surprisingly, these numbers increased to 845 (cluster A; 48.2% of the 1752 pairs) and 1258 (cluster G; 71.8%) pairs of co-expressed genes upon analysis of the sNucRNA-seq datasets, for a total of 1664 pairs of genes (94.9%) expressed in at least one cluster (Figure 2B). The difference in these results between single-protoplast- and single-nucleus-based RNA-seq analyses is unexpected, considering that scRNA-seq technology captures more transcripts per biological entity than sNucRNA-seq technology [29]. We hypothesize that the relative abundance of membrane-associated protein-encoding transcripts is lower in the cellular transcriptome versus the nuclear transcriptome, leading to their limited detection. Further analyses of the quality of the libraries and the saturation of their sequencing should be conducted to verify this hypothesis. Biologically, we hypothesize that the genes encoding interacting protein membranes are co-expressed, as reflected by the higher correlation from the sNucRNA-seq datasets. In the cell, the differential half-life of the transcripts might lead to an overall decrease in these correlation due to changes in the relative abundance of transcripts encoding interacting membrane proteins.

Nevertheless, looking at the level of expression of the two genes composing a pair in our sc and sNucRNA-seq analyses, our analysis revealed correlations between the expression of the two genes forming a pair from both the sNucRNA-seq and scRNA-seq datasets (coefficient correlations ranging from 0.17 (cluster 7) to 0.74 (cluster 1) upon analysis of the scRNA-seq datasets (Figure 3A); and from 0.385 (cluster L) to 0.69 (cluster F) upon analysis of the sNucRNA-seq datasets (Figure 3B)). To estimate the significance of these findings, we conducted a similar analysis on 1752 randomly selected gene pairs. The coefficient correlations were lower compared to our analysis of the membrane protein gene pairs (except for three outliners (i.e., clusters 8, 9, and 14), the coefficients of correlations ranged from −0.08 (cluster 18) to 0.09 (cluster 13) upon analysis of the scRNA-seq datasets (Appendix A); and from 0.009 (cluster E) to 0.18 (cluster D) upon analysis of the sNucRNA-seq datasets (Appendix A)). Taken together, our results support that genes encoding interacting membrane proteins are co-regulated at the single cell level, likely to enable the formation of cell-type-specific protein complexes in the membrane. Such complexes differ between Arabidopsis root cell types to reflect their unique biological needs.

### 3.3. The Membrane Proteome of the Arabidopsis Root Cells Differs between Cell Types

Hypothesizing that the membrane proteome differs between Arabidopsis root cells and cell types, we conducted a clustering analysis of the Arabidopsis protoplasts and nuclei based on the transcriptional activity of 965 membrane protein-encoding genes [6] (Appendix A). Upon clustering, the 6474 protoplasts and 8880 nuclei isolated from the Arabidopsis root were distributed in 8 and 5 clusters, respectively (Figure 3A,B). The medians of 519 and 135 membrane protein-encoding genes were found, expressed per protoplast and nucleus, leading to the identification of 962 (99.68%) and 964 (99.89%) membrane protein-encoding genes in the Arabidopsis root. To annotate these clusters, we identified 28 and 19 genes encoding membrane proteins which were preferentially expressed in each of the 8 and 5 protoplast and nuclei clusters, respectively (Figure 3C,D, left panels). Mining our annotated whole scRNA- and sNucRNA-seq UMAPs (Figure 1A,B), we observed the cell-type preferential activity of the 28 and 19 selected membrane protein-encoding genes in a limited number of clusters (Figure 3C,D, right panels). These transcriptional patterns supported the annotation of the 8 and 5 root protoplasts and nuclei in the four groups reflecting the major cell types composing the Arabidopsis root (i.e., epidermal, cortical, endodermal, and stele cells; Figure 3A,B). 

Taken together, our results highlight that the different cell types composing the Arabidopsis root are characterized by a unique transcriptomic profile of the membrane protein-encoding genes. These results suggest that the membrane proteome differs between Arabidopsis root cell types, likely to reflect the unique biological function of the cells and their interactions with the surrounding cells. We assume that the unique composition of the membrane proteome is also playing a critical role in inducing differential responses between cell types.

## 4. Conclusions

Plant single-cell biology is emerging, notably through the use of microfluidic technologies. The first use of these technologies led to the establishment of the transcriptome of the Arabidopsis root upon analysis of individual protoplasts [10]. In this study, for the first time, we shared the use of isolated nuclei to generate meaningful transcriptomic information and to reveal the differential profiles of chromatin accessibility at the single cell level [28]. Efforts are currently being made to conduct single-cell/cell-type proteomic and metabolomic studies [32,33].

These analyses revealed the differential use of plant genomes and genes, and the differential abundance of proteins and metabolites between plant cells and cell types. Here, to support the differential composition of the plasma membrane proteome and previously characterized interactions between membrane proteins [6], we report the cell-type-specific activity of the Arabidopsis root genes encoding membrane proteins. Our results suggest that the membrane proteome differs for each cell type composing the Arabidopsis root. Considering the central role of the plant plasma membrane in plant cell development and the plant cell response to environmental stresses [34,35], our study is a first step toward understanding the role of the plasma membrane proteins in controlling the biology of various plant cell types, including the regulation of biochemical pathways and genetic programs in response to environmental signals.

## Figures and Tables

**Figure 1 membranes-12-00874-f001:**
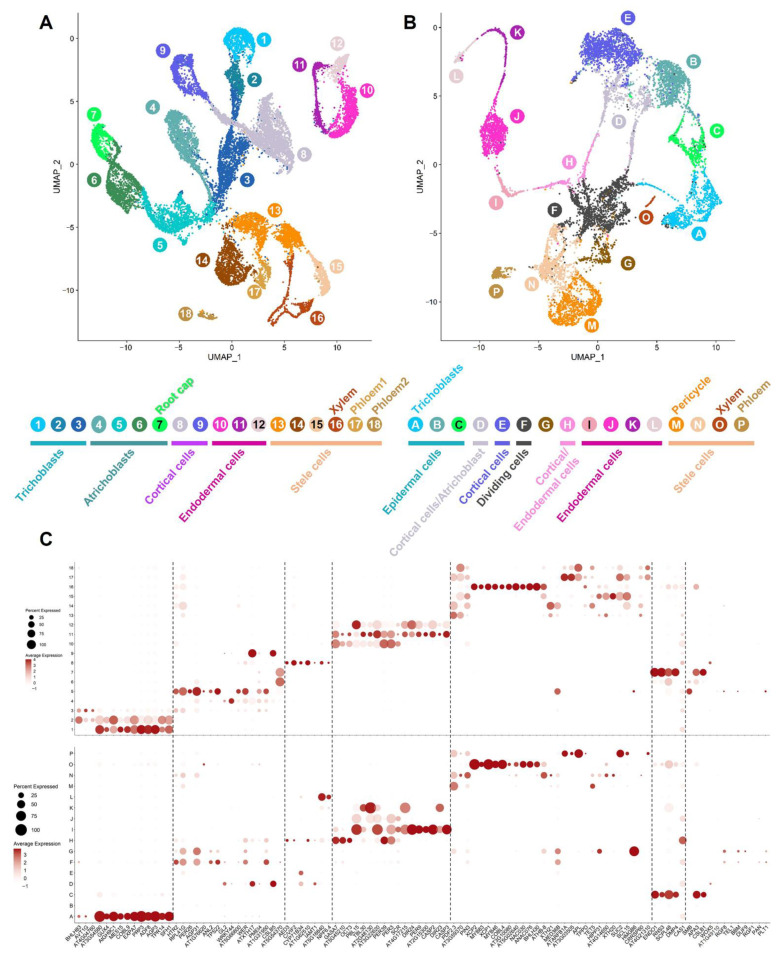
Functional annotation of the *Arabidopsis* root cell types upon reanalysis of the scRNA-seq and sNucRNA-seq datasets [28]. The transcriptional pattern of 101 Arabidopsis root cell-type and cell-death marker genes was used to assign specific root cell types upon analysis of scRNA-seq ((**A**), clusters 1 to 18) and sNucRNA-seq datasets ((**B**), clusters A to P). The normalized expression levels of these marker genes are provided in the context of the 18 protoplasts (top panel) and 16 nuclei Arabidopsis root clusters (bottom panel), respectively (**C**). The percentage of nuclei expressing the gene of interest (circle size) and the mean expression (circle color) of the genes are shown.

**Figure 2 membranes-12-00874-f002:**
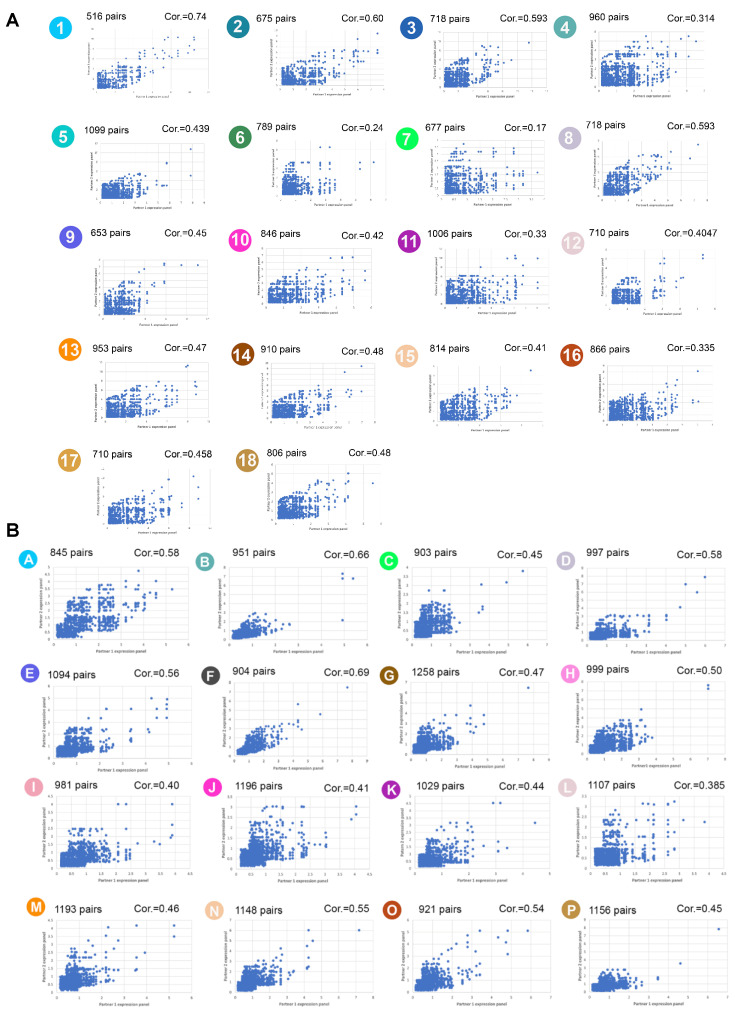
Correlation analysis of the expression of Arabidopsis genes encoding for interacting membrane-associate proteins. For each protoplast- ((**A**), clusters 1 to 18) and nuclei-based ((**B**), clusters A to P) cluster, the average numbers of UMIs per protoplast/nuclei of each gene composing the pair are indicated on the x-axis and y-axis. For each cluster, The total number of pairs with both genes expressed is indicated, as well as the coefficient of correlation between the expression of the two genes.

**Figure 3 membranes-12-00874-f003:**
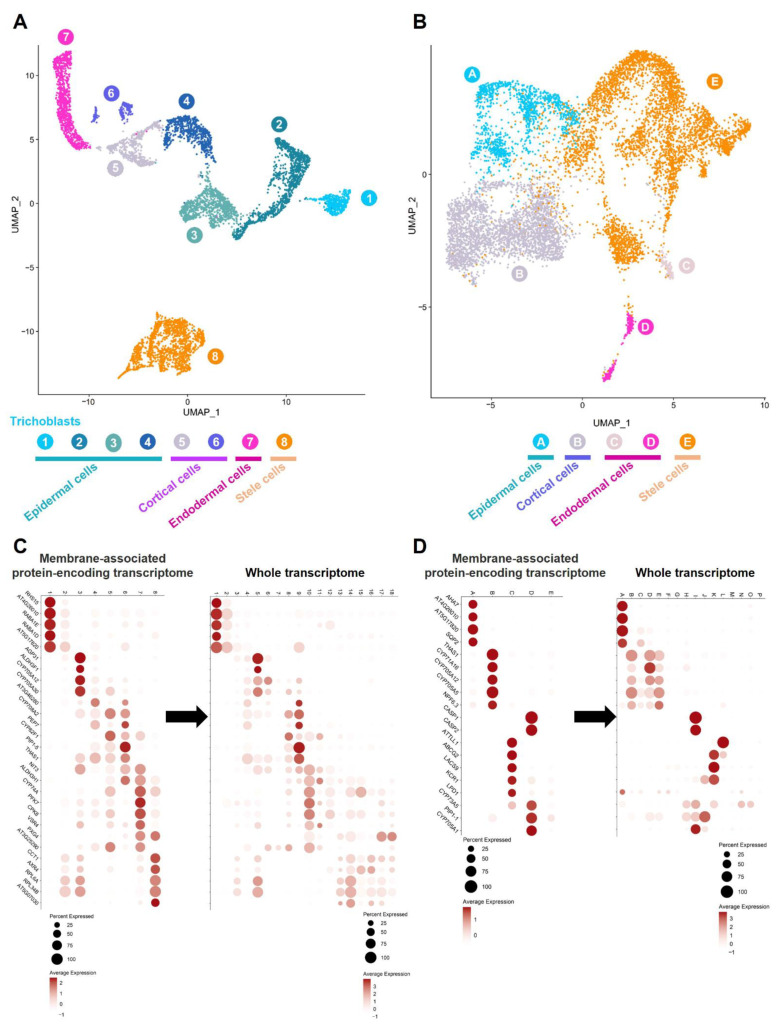
Functional annotation of the Arabidopsis root cell types according to the activity of 965 membrane protein-encoding genes. Both scRNA-seq (**A**) and sNucRNA-seq UMAPs (**B**) were independently annotated. The annotation of the UMAPs was conducted by identifying protoplasts (**C**) and nuclei (**D**) cluster-preferentially expressed membrane-associated protein-encoding genes (Figure 1C), then questioning their activity in the whole transcriptome scRNA-seq and sNucRNA-seq datasets (i.e., Figure 1C). The percentage of nuclei expressing the gene of interest (circle size) and the mean expression (circle color) of the genes are shown.

## Data Availability

The transcriptomic datasets used in this study come from the query datasets from National Center for Biotechnology Information (NCBI) with the accession: GSE155304 and the samples with the accessions GSM4698755, GSM4698756, GSM4698757, GSM4698758, GSM4698759, and the accession number GSE123013 (samples: GSM3490689, GSM3490690, GSM3490691), the processed data presented in this study is available on request from the corresponding author.

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
