# Peer review of "Cell-Type-Specific Profiling of the *Arabidopsis thaliana* Membrane Protein-Encoding Genes"

_membranes, 2022, doi:10.3390/membranes12090874_

Round 1
Reviewer 1 Report
Dear Authors,
The manuscript has lots of applications in plant research, and how proteins/metabolites interact due to biotic and abiotic stress.
My only request to the authors to highlight/include which are the proteins encoding genes identified in your study and what are their function/role in membranes.
Regards
Reviewer
Author Response
We added two supplemental tables to better support the genes used in our analysis. When generating these files, we notice a mistake with one number reported in the main manuscript. This has been modified accordingly.
Reviewer 2 Report
Dear the editor, Libault and Perez submitted a manuscript titled "Cell-type-specific profiling of the Arabidopsis thaliana membrane protein-encoding genes". This manuscript is of great novelity, and made of my day.
As we are all known that membrane proteins play important roles in transduction external signals and response to the cellular environment. They have performed a very comprehensive analysis of the expression of Arabidopsis membrane proteins at single cell resolution. As they have said, they established a linkage between previously reported interactions and co-expression of genes at the single cell-type resolution.
After I reviewed this manuscript, I feel that whole sections are well-written. Especially, I am very impressive about the figures. I do not find any flaws in this manuscript. Thus I recomend accepted in current form. Thanks!
Author Response
We want to thank Reviewer #2 for these very positive comments.
